# Development of a Scale to Measure Infant Eating Behaviour Worldwide

**DOI:** 10.3390/nu13082495

**Published:** 2021-07-22

**Authors:** Charlotte M. Wright, Jessica Megan Gurney, Antonina N. Mutoro, Claudia Shum, Amara Khan, Beatrice Milligan, Widya Indriani, Loukia Georgiou, Stephanie Chambers, Rachel Bryant-Waugh, Ada L. Garcia

**Affiliations:** 1Child Health, School of Medicine, Dentistry and Nursing, College of Medical, Veterinary and Life Sciences, University of Glasgow, Glasgow G51 4TF, UK; 2Human Nutrition, School of Medicine, Dentistry and Nursing, College of Medical, Veterinary and Life Sciences, University of Glasgow, Glasgow G31 2ER, UK; jmgurney@doctors.org.uk (J.M.G.); antoninanamaemba@gmail.com (A.N.M.); chsshum@gmail.com (C.S.); amarak830@gmail.com (A.K.); 2172596M@student.gla.ac.uk (B.M.); widyaindria@yahoo.co.id (W.I.); lucygeorgiou3011@gmail.com (L.G.); Ada.Garcia@glasgow.ac.uk (A.L.G.); 3African Population and Health Research Centre, Nairobi P.O. Box 10787-00100, Kenya; 4School of Social and Political Sciences, University of Glasgow, Glasgow G12 8RS, UK; Stephanie.Chambers@glasgow.ac.uk; 5Maudsley Centre for Child and Adolescent Eating Disorders, South London and Maudsley NHS Foundation Trust, London SE5 8AZ, UK; rachel.bryant-waugh@slam.nhs.uk

**Keywords:** undernutrition, global nutrition/health, feeding problems, eating behaviour, appetite, complementary feeding

## Abstract

In order to create a short, internationally valid scale to assess eating behaviour (EB) in young children at risk of undernutrition, we refined 15 phrases describing avidity or food refusal (avoidance). In study one, 149 parents matched phrases in English, Urdu, Cantonese, Indonesian or Greek to videos showing avidity and avoidance; 82–100% showed perfect agreement for the avidity phrases and 73–91% for the avoidant phrases. In study two, 575 parents in the UK, Cyprus and Indonesia (healthy) and in Kenya, Pakistan and Guatemala (healthy and undernourished) rated their 6–24 months old children using the same phrases. Internal consistency (Cronbach’s α) was high for avidity (0.88) and moderate for avoidance (0.72). The best-performing 11 items were entered into a principal components analysis and the two scales loaded separately onto 2 factors with Eigen values > 1. The avidity score was positively associated with weight (r = 0.15 *p =* 0.001) and body mass index (BMI) Z scores (r = 0.16 *p =* 0.001). Both high and low avoidance were associated with lower weight and BMI Z scores. These scales are internationally valid, relate to nutritional status and can be used to inform causes and treatments of undernutrition worldwide.

## 1. Introduction

Eating and feeding behaviour (EB) problems in preschool children are common [1,2], cause great anxiety [3] and may also cause or complicate undernutrition [4,5]. Studies in low- and middle-income countries (LMIC) have observed higher refusal of food in stunted growth compared to healthy infants [6] and reduced appetite associated with common illnesses, leading to lower food intake [7]. Food refusal in some children can be attributed to an inherently lower drive to eat [8], appetite suppression caused by illness or stress [7], unresponsive feeding styles [9] or a combination of these factors; thus, assessing EB is important, but there are few suitable tools to assess these in young children [10] who are most at risk of undernutrition.

The World Health Organization (WHO) treatment protocol for severe acute malnutrition (SAM) includes an ‘appetite test’, an observation of the child’s immediate willingness to take a prescribed food supplement. Direct observation is labour-intensive and may be unrepresentative [11], but there is no other standard means of assessing eating behaviour in the LMIC context [12]. Parental-report-based measures of feeding and eating problems have been developed for use in more affluent clinic settings, although these do not relate specifically to undernutrition and are mainly aimed at older children [13]. 

The Child Eating Behaviour Questionnaire (CEBQ) is the most widely used and validated EB scale worldwide, although this was initially developed for older children [14], focusses mainly on risk factors for obesity [15] and is too long for use in an LMIC clinical setting. A single question about appetite in the infancy version of the CEBQ correlated with all its subscales [15] and generally predicted weight gain better than other subscales [16]. In the UK Gateshead Millennium cohort (GMS), a score combining descriptions of various food refusal behaviours was negatively associated with weight gain, while a single rating of appetite was the strongest positive predictor of weight gain [9]. However, the word ‘appetite’ does not have a direct translation in most other languages and its associations vary with age [9]. Disappointingly, most of the other terms selected to describe enthusiasm for eating in the GMS proved ambiguous [17]. A subsequent study in Nairobi using the GMS phrases found that undernourished infants showed increased food refusal and lower interest in food compared to healthy peers [18], although again most of the items selected to assess avidity were not consistently inter-correlated or associated with weight gain. The aim of this program of work was, therefore, to develop a short measure of EB that would be robust to translation, would assess both positive enjoyment of food (avidity) and food refusal (avoidance) and would be valid for use in the assessment of young children at risk of undernutrition worldwide. The objectives of this study are to: (1)Identify and refine candidate phrases expressed in simple language; translate these into a range of languages; and assess their face validity (study one), internal consistency and predictive validity (study two);(2)Combine the best-performing phrases into scores for avidity and avoidance and assess their predictive validity in terms of infant weight gain (study two)

## 2. Materials and Methods

The sequencing of the two studies is shown in Appendix A.

### 2.1. Selecting and Developing the Candidate Phrases

Phrases known to be related to either avidity or avoidance from our previous studies (*n* = 6) [18] and from the CEBQ (*n* = 9) [14] were identified. These phrases were first tested in Glasgow, UK, in English and Urdu, which is widely spoken by Pakistani-origin families in Scotland. They were translated into Urdu by two native speakers and then back-translated by two different Urdu speakers. The best combinations of English and Urdu phrases (1st prototype) were then selected for use in a pilot study with English- and Urdu-speaking UK mothers and grandmothers using a video-based procedure (see below). The results were discussed by the research team and a 2nd prototype created (Appendix A).

### 2.2. Study One

The face validity of the 2nd prototype was tested in English and Urdu in the UK, in Cantonese in Hong Kong, in Greek in Cyprus and in Indonesian in Java. The choice of languages was based on convenience in terms of native speakers in the research team and access to participants. The phrases were re-translated into Urdu, Cantonese, Greek and Indonesian, each by a single native speaker, then back-translated by two other native speakers to identify and resolve any ambiguities. In most languages, the word for ‘love’ would not be applied to foods, so the phrase ‘loves food’ was usually translated as ‘likes food a lot’.

#### 2.2.1. Participants

Depending on the setting, these were either adults who had ever parented a child aged more than 6 months (UK and Hong Kong) or parents of a child aged 6 months to 2 years (Cyprus and Indonesia). The sampling details for each centre and ethical approval are shown in Appendix A.

#### 2.2.2. Video-Based Validity Test Procedure

In order to test the phrases without the use of any potentially indicative language, short sections from four videos were prepared, all with parental consent (Appendix A). Two showed healthy children eating avidly, with little or no avoidance (enthusiastic), with one being recorded for an earlier study [19] and the other recorded specially for this study. Two others showed children with food refusal (FR) recorded for therapeutic purposes. The videos were then used in interviews undertaken by a fluent speaker of the relevant language. All researchers were trained in the test procedure. Participants were given the list of phrases and asked to watch each video in turn. They were then asked to rate the extent to which each phrase matched the child in the video on a 5-point scale (1 = very likely, 2 = likely, 3 = uncertain, 4 = unlikely, 5 = very unlikely). Even if a specified behaviour was not shown in the video, respondents were asked to rate the likelihood that a child like this might show this behaviour. 

#### 2.2.3. Analysis

The results were analysed using SPSS Version 26. We hypothesised that the avid phrases would be rated as most applicable to the enthusiastic videos and least applicable to the FR videos, and vice versa for the avoidant phrases. We, therefore, created an overall agreement score per respondent, combining all their responses to each avidity phrase: 

= (6 − average for 2 enthusiastic videos) + average for 2 FR videos.

Similarly, for the avoidant phrases:

= average for 2 enthusiastic videos + (6 − average for 2 FR).

These generated scores with a minimum of 2 and a maximum of 10. For both scales, perfect agreement with a phrase would yield a score of 8 or more, while uncertainty would yield a score of between 6 and 8 and lack of agreement would yield a score of less than 6. We also took the average of all the individual phrase agreement scores to arrive at an overall agreement score for each phrase per individual.

### 2.3. Study Two

Primary carers in 6 countries were asked to rate the applicability of each candidate phrase to their own child aged 6–24 months.

#### 2.3.1. Participants

Full details of the different samples and ethical approvals are shown in Appendix A. In Cyprus and Indonesia, the survey questions were asked in combination with study one (see above), with only healthy children included. In Kenya, Pakistan and Guatemala, both healthy (weight for length (WLZ) > −2SD and a mid-upper-arm circumference (MUAC) ≥ 12.5 cm) and undernourished (WLZ ≤ −2SD or MUAC < 12.5 cm) children were recruited. Children who required inpatient care or with congenital disorders or disabilities were excluded.

#### 2.3.2. Procedure

The phrases were translated by a single native speaker and back-translated by two other native speakers to identify and resolve any ambiguities, then incorporated into a questionnaire or interview schedule. All parents consented in person and then were interviewed or given a questionnaire to complete. Child height and weight were also measured in Pakistan, UK, Kenya, and Guatemala using WHO guidelines [20].

#### 2.3.3. Analysis

The weight, length, and body mass index (weight (kg)/height (m)^2^ values were converted into standard deviation (z) scores compared to the WHO growth standard [21] using LMS growth [22]. The correlations of the individual avidity and avoidance items with each other and with weight and BMI Z scores were explored using Spearman’s correlation to avoid assumptions of linearity. Cronbach’s alpha (Cα) values were calculated for all the items in each scale and with each item excluded. Once items with low consistency, validity or repetition were excluded, the remaining items were entered together into a principle components analysis (PCA) to assess the extent to which the avidity and avoidance items loaded together. Scores for avidity and avoidance were then constructed by taking the average for all included items, yielding scores ranging from 1 for low to 5 for high avidity or avoidance. These scores were then divided into tertiles to examine possible non-linearity in association with weight and BMI Z scores, using ANOVA with or without a trend. 

### 2.4. Sample Size

For both studies, successive samples in different settings were collected over time, so that power calculations for the total pooled samples were not undertaken in advance; however, post hoc calculations for study one (using EpiInfo statcalc Version 7.2.3.1) suggested that with 6 clusters of around 20, we had 80% power to estimate agreement levels for any one phrase to within ±3.5% but only sufficient power to detect quite large differences between individual language groups (e.g., 60% versus 95% agreement). For study two, each centre aimed to recruit at least 50 healthy children, and where these were included, 50 malnourished children; using the Altmann normogram [23] suggested that this would give 80% power at the 95% level to detect a mean difference in scores of a clinically important difference of just over half a standard deviation between each setting or group. 

## 3. Results

### 3.1. Study One

Study one involved 149 participants, of whom 22 spoke English, 18 Urdu, 32 Cantonese, 50 Indonesian and 27 Greek. Overall, there was very high agreement between all the phrases and the videos (Table 1, Appendix A). Participants showed perfect agreement of 87–100% for the avidity and 73–91% for the avoidant phrases. Very few found any of the phrases ambiguous (i.e., matching them to the opposite of the expected video), but up to 26% were uncertain about matching individual phrases, particularly those related to avoidance (Table 1).

The avidity phrases showed a number of significant inter-language differences (Table 2), mainly reflecting the differences between the extent of agreement, rather than differing levels of disagreement, as there were only 3 instances where less than 85% of participants showed perfect agreement with any one phrase in any language. Two of these phrases, ‘always asking for food’ and ‘enjoys a wide variety’, showed lower agreement in the English speakers, while the Indonesian participants showed lower agreement with ‘willing to try new foods’. For the avoidant phrases, there was less variation between languages, although the native English speakers were less likely to show perfect agreement, and for two phrases only around 50% showed perfect agreement. One of these, ‘holds food in mouth’, also showed low agreement in two other language groups. A meal length of 30 min also showed widely varying agreement, although there was more agreement about meals lasting an hour.

### 3.2. Study Two

Data on eating behaviour were collected for 573 children (UK = 107, Pakistan = 108, Kenya = 157, Guatemala = 125, Cyprus = 27, Indonesia = 50). The mean age was 14.2 (SD, 5.4) months (50.4% female). Of these, 478 also had growth data and 29% of these children had weight Z scores below −2SD (Appendix A). In Kenya, 19 children were seen twice at a mean interval (SD) of 3.3 (0.98) weeks apart.

The avidity items showed generally high internal consistency (overall Cronbach’s alpha (CA) = 0.88), which was similar between countries (CA 0.84–0.91), except for Guatemala (CA = 0.59) (Table 3). All but one item had at least borderline associations with weight or BMI Z scores (Table 4). Avoidance showed moderate internal consistency, with an overall CA of 0.72, again with similar internal consistency between most countries (CA = 0.71–0.77), except for Guatemala (CA = 0.60) (Table 3). All items had at least borderline associations with weight or BMI Z scores (Table 4). 

### 3.3. Creating Avidity and Avoidance Scores

The items ‘eats quickly’ and ‘always asking for food’ tended to show weaker correlation with other items and the CA rose slightly when these were excluded (Table 3). However, ‘eats quickly’ was significantly associated with both weight and BMI, while ‘always asking for food’ was not. As such, ‘always asking for food’ and ‘holds food in mouth’ were not included in the final scale, along with ‘willing to try new foods’, which was similar to the two other items and was uncorrelated to weight or BMI. The two items related to meal length showed the weakest internal correlation. When either was excluded, the CA rose slightly (Table 3). However, it was judged important to include one rating of meal length, so ‘meals last more than an hour’ was selected, as in study one it showed higher face validity and less inter-country variation.

When these items were entered into the PCA, this yielded only two factors with Eigen values > 1. The first factor loaded all of the avidity variables, all but one R > 0.6, and explained 41% of total variance. The second factor explained 15% variance and 5/6 of the avoidance variables loaded R > 0.47. There was also a third factor with Eigen value = 0.998, which loaded only ‘meals last for >1 h’ (r = 0.94), which explained another 9% of the variance. This provided further justification to include that item.

After these exclusions, scores for avidity and avoidance were constructed. For the total sample the median [25,75] avidity score was 3.7 [2.7–4.3] and median avoidance score was 2.0 [1.6–2.6]. The two scores were uncorrelated with age and moderately correlated with each other (Spearman’s R (SR) = 0.41 *p <* 0.001). The strength of correlations between the scores varied by country, being strongest for the UK (SR = 0.76) and weakest for Pakistan (0.45) and Guatemala (0.42). 

### 3.4. Predictive Validity

Avidity was positively associated and avoidance was negatively associated with weight and BMI Z scores, but those with both low and high avoidance had significantly lower weight and BMI values (Table 5). The extent and direction of correlation of the scores with weight and BMI varied by country. Kenyan children showed significant linear associations of both scores with weight and of avidity with BMI. Pakistani children with both low and high Avoidance had significantly lower weight and BMI, but there was no association of Avidity with weight or BMI. No UK children were undernourished and no associations were found for them between either weight or BMI with either score, while Guatemalan children only showed borderline associations of BMI with both scores (Table 5).

## 4. Discussion

In order to prevent and manage child malnutrition, a wide range of risk factors need to be addressed, including care factors such as responsive feeding [24]. However, studies of the child’s role in the feeding interaction have been handicapped until now by a lack of valid assessment tools [18]. This program of work aimed to develop a short measure of EB that would be valid for use worldwide. It benefited from the involvement of researchers from a wide range of countries and language groups, with large samples drawn from seven different countries. In the first study, we found high agreement between most phrases and the videos in all settings, suggesting that caregivers accurately identify typical avidity and avoidant behaviours in young children. In the second study, we showed that most of the different items relevant to avidity and avoidance showed moderate to strong internal consistency, while, after exclusion of the weaker items, the resulting avidity and avoidance scores were associated with child weight and BMI.

Child malnutrition remains a worldwide problem [24] and any measure designed to contribute to its assessment needs to be applicable across a wide range of languages and settings. The usual approach to the development of such tools is to construct and validate the measure in one language first, then translate them into other languages where they then have to be revalidated. Apart from the labour involved in doing this, if the language originally used translates poorly, the scale cannot then be changed and the scale may then only be useable in one setting. For example, a French feeding difficulty questionnaire [25] was recently translated into English [26], where only some of the original items proved valid; thus, we opted to test the tool in translation from the outset.

The best-established measure for the assessment of childhood eating behaviour to date, the CEBQ, was developed [14] and validated in English [27], and subsequently has been widely translated and used worldwide. However, the CEBQ is too long to be applicable in less affluent field settings and it has predominantly been validated in the context of over- rather than undernutrition. Nonetheless, the 35 simply worded CEBQ items include many phrases of relevance to enjoyment of food, which provided an invaluable pool of phrases to test. Ultimately, all 6 of the final avidity score phrases were based on CEBQ items, although 3 had been modified and only 3 came from the enjoyment of food subscale.

### 4.1. Strengths

A novel feature of the first study was the assessment of face validity via matching of the phrases to children observed in videos, which avoided using other words that could otherwise have influenced parental responses. A further strength was that in the three LMIC centres, we were able to oversample malnourished children, which increased the level of variability and our power to detect associations with weight and BMI.

### 4.2. Limitations

The videos could not show every described behaviour, although respondents were asked to rate the likelihood that a child ‘like this’ might show this behaviour. There was a risk that those behaviours that were not shown in the video would most often be rated as ‘uncertain’. The children in the videos were all white British, and future work using a wider range of video subjects from different ethnic groups is needed. The researchers conducting the interviews were not blind to which videos and phrases were hypothesised to be related to avidity or aversion. However, they were trained to avoid coaching or suggesting responses and in most centres parents rated the videos using self-completion forms. In some centres, older parents of grown-up children also took part, who might have had different views in retrospect. While study one was powered to estimate overall agreement scores quite precisely, it was less powered to detect differences between language groups, although numbers were sufficient to detect the more substantial inter-language variations seen for the less effective phrases evaluated. Finally, in study two, the between-country analyses were limited by the relatively small sample sizes and the variable proportions of malnourished children.

Overall, there were inverse linear associations between avidity and both weight and BMI; the association was strongest for weight, which appeared to have a tendency for low avidity in the underweight, rather than high avidity in the overweight. In contrast, examination by tertile of the avoidance scale revealed non-linear relationships with both high and low avoidance associated with lower weight and BMI, with considerable variation between countries. This could suggest that a degree of food refusal is healthy and that children in some settings may become undernourished as a result of high levels of avoidant eating behaviour, while others become apathetic as a result of undernutrition and display less avoidance. Because of the large number of centres, there is a risk of chance findings due to multiple significance testing, so any apparent inter-country differences must be treated with caution. Future work could consider whether EB in malnourished children may in fact be different or be perceived differently in South Asia compared to Africa or South America.

Remarkably few studies have assessed the extent to which eating behaviour predicts nutritional status in the context of malnutrition. Observational studies have found undereating in stunted children [6] and that moderately malnourished children did not finish meals [28]. However, assessment via observation is highly labour-intensive [11], but two previous studies have shown that caregiver reports predict observed behaviour [7,28], suggesting that they are a valid alternative. As far as we know, ours is the first scale to be applied in an LMIC setting that has been shown to have predictive validity in terms of weight gain. A scale has been developed for use in children at risk of malnutrition in Bangladesh, but this has so far not been shown to predict either dietary intake or anthropometry [12].

### 4.3. The Final Eating Behaviour Scores

The 11 best-performing items were combined into two scores for avidity (comprising child likes food a lot, is interested in food, enjoys eating, enjoys a wide variety of foods, eats quickly and finishes meals) and avoidance (comprising turns head when offered food, pushes food away, cries and screams during meals, spits out food and meals last more than one hour). These have now been incorporated into a broader assessment tool, the International Complementary Feeding Evaluation Tool, which is currently being tested in a range of countries [29].

In a clinical setting, these scales could be used to identify and monitor behavioural feeding problems to allow more targeted interventions. In a research setting, these will allow the comparison of eating behaviours between and within countries and contribute to a better understanding of the roles avidity and avoidance play in causing or perpetuating acute undernutrition. Further work is needed to explore the extent to which these scores vary in clinical populations, change over time or may be affected by different interventions.

## 5. Conclusions

We have identified descriptions of infants enjoying or avoiding food that are widely recognised, associated with nutritional status and show consistent inter-correlations in diverse languages and settings. We propose that these can be used to measure eating behaviour worldwide.

## Figures and Tables

**Table 1 nutrients-13-02495-t001:** Median agreements scores for individual phrases in study one, as applied to the two enthusiastic eating and two food refusal videos and percentage of participants showing full agreement, uncertainty and disagreement for each phrase across the four videos.

	Rating of Match to Video Type Median (Range)		Overall Agreement Scores across the Four Videos % (N)		
	Enthusiastic	Food Refusal	Full	Uncertain	Disagree
**Avidity Phrases:**	1 = very likely, 5 = very unlikely		≥8	6 to <8	<6
Loves food	1.0 (1–3)	4.5 (2.5–5)	97 (144)	3 (4)	0
Enjoys eating (*n* = 114)	1.0 (1–2.5)	5.0 (3–5)	100 (114)	0	0
Is interested in food	1.0 (1–3)	4.5 (2–5)	97 (143)	3 (5)	0
Is always asking for food	1.0 (1–4.5)	4.5 (2–5)	88 (130)	11 (16)	1 (1)
Enjoys a wide variety of foods	1.25 (1–3)	4.5 (2–5)	87 (129)	13 (19)	0
Eats quickly	1.0 (1–3.5)	5.0 (2–5)	98 (145)	2 (3)	0
Is willing to try new foods	1.0 (1–3)	5.0 (1–5)	82 (122)	18 (26)	0
Finishes his/her meal	1.0 (1–3)	4.5 (1.5–5)	94 (137)	5 (8)	1 (1)
**Avoidant phrases**					
Turns head away	5 (3–5)	1.5 (1–4.5)	90 (134)	10 (14)	0
Pushes food away	5 (3–5)	1.5 (1–4.5)	91 (135)	7 (11)	1 (2)
Cries/screams	5 (2–5)	2 (1–4.5)	82 (121)	16 (24)	2 (3)
Holds food in mouth for too long	5 (2–5)	2 (1–5.0)	73 (108)	26 (38)	1 (1)
Spits out food	5 (2–5)	2 (1–4.0)	86 (127)	14 (20)	1 (1)
Meals often last more than 30 min	5 (1–5)	1.5 (1–5)	81 (120)	14 (20)	5 (8)
Meals sometimes last more than an hour	5 (1–5)	1.5 (1–5)	85 (126)	13 (19)	2 (3)

**Table 2 nutrients-13-02495-t002:** Percentages of participants showing full agreement with each phrase across the four videos in study one, broken down by language group.

Values Are Overall % Agreement (N)						
Avidity Phrases:	English	Urdu	Cantonese	Greek	Indonesian	P χ^2^
Loves food	86% (18)	100% (18)	97% (31)	100% (27)	100% (50)	0.01
Enjoys eating *	100% (5)	-	100% (32)	100% (27)	100% (50)	NS
Is interested in food	86% (18)	100% (18)	100% (32)	100% (27)	96% (2)	0.03
Is always asking for food	60% (12)	100% (18)	88% (28)	96% (26)	92% (46)	0.003
Enjoys a wide variety of foods	67% (14)	100% (18)	91% (29)	93% (25)	86% (43)	0.02
Eats quickly	95% (20)	94% (18)	97% (31)	100% (27)	100% (50)	NS
Is willing to try new foods	62% (19)	100% (18)	84% (30)	100% (27)	74% (37)	0.001
Finishes his/her meal	86% (18)	100% (18)	97% (31)	93% (25)	94% (47)	NS
**Avoidant phrases**						
Turns head away	86% (18)	83% (15)	94% (30)	100% (27)	88% (44)	NS
Pushes food away	90% (19)	83% (15)	97% (31)	100% (27)	86% (43)	NS
Cries/screams	81% (17)	89% (16)	81% (26)	85% (23)	78% (39)	NS
Holds food in mouth	52% (11)	78% (14)	53% (17)	89% (24)	84% (42)	0.008
Spits out food	71% (15)	94% (17)	84% (27)	100% (27)	82% (41)	0.07
Meals often last > 30 min	48% (10)	94% (17)	91% (29)	93% (24)	78% (34)	<0.001
Meals sometimes last > hour	48% (15)	94% (17)	91% (30)	96% (25)	88% (39)	<0.001

* In the early English and all Urdu versions, this was missed out in error.

**Table 3 nutrients-13-02495-t003:** Spearman’s correlations for different candidate items and effects on Cronbach’s alpha if individual items are deleted for avidity and avoidance scores in study two. Note: *p* all < 0.001 unless stated.

Avidity	Interested in Food	Enjoys Eating	Always Asking for Food	Enjoys Wide Variety of Foods	Eats Quickly	Willing to Try New Foods	Finishes Meal	Cα ^2^
Loves food	0.82	0.81	0.32	0.45	0.26	0.50	0.54	0.864
Interested in food		0.84	0.36	0.50	0.28	0.58	0.61	0.857
Enjoys eating			0.38	0.56	0.28	0.59	0.60	0.855
Always asking for food ^1^				0.36	0.46	0.27	0.41	0.883
Enjoys a wide variety of foods					0.41	0.55	0.44	0.865
Eats quickly						0.28	0.37	0.885
Willing to try new foods							0.50	0.866
Finishes meal								0.863
Cronbach’s alpha when all included								0.880
**Avoidance**	**Pushes Food Away**	**Cries/Screams during Meals**	**Holds Food in Mouth**	**Spits out Food**	**Meals Last** **>30 min**	**Meals Last** **>1 h**	**Cα ^2^**	
Turns head when offered food	0.65	0.46	0.35	0.42	0.13 ^3^	0.17	0.65	
Pushes food away		0.49	0.32	0.46	0.13 ^3^	0.18	0.64	
Cries and screams during meals			0.39	0.40	0.19	0.10 ^4^	0.658	
Holds food in mouth				0.37	0.12 ^3^	0.22	0.683	
Spits out food					0.11 ^3^	0.17	0.668	
Meals last more than 30 min						0.19	0.751	
Meals last more than an hour							0.738	

^1^ Missing in Kenya; ^2^ Cronbach’s alpha if item deleted; ^3^
*p* < 0.01; ^4^
*p* < 0.05.

**Table 4 nutrients-13-02495-t004:** Spearman’s correlations with weight and BMI Z scores of individual phrases and for avidity and avoidance scores in study two.

	Weight Z Score		BMI Z Score	
	R ^1^	*p* ^2^	R ^1^	*p* ^2^
Child loves food	−0.02	0.61	−0.08	0.09
Child is interested in food	−0.04	0.44	−0.10	0.03
Enjoys eating	−0.10	0.04	−0.13	0.004
Always asking for food	−0.05	0.38	−0.05	0.41
Enjoys a wide variety of foods	−0.21	<0.001	−0.17	<0.001
Eats quickly	−0.23	<0.001	−0.13	0.004
Willing to try new foods	−0.04	0.34	−0.06	0.23
Finishes meal	−0.06	0.17	−0.10	0.04
Turns head when offered food	0.01	0.79	0.09	0.04
Pushes food away	0.08	0.08	0.01	0.03
Cries and creams during meals	0.14	0.002	0.16	0.001
Holds food in mouth	0.05	0.32	0.11	0.02
Spits out food	−0.08	0.08	−0.03	0.55
Meals last > 30 min	0.11	0.02	0.08	0.07
Meals last > 1 h	0.09	0.06	0.03	0.53
Avidity score	0.15	0.001	0.16	0.001
Avoidance score	−0.09	0.046	−0.12	0.007

^1^ Spearman’s correlations; ^2^ Significance of R.

**Table 5 nutrients-13-02495-t005:** Predictive validity for whole sample and per country in study two, showing differences in mean weight and BMI by tertile for avidity and avoidance scores.

	Total		Kenya		Pakistan		Guatemala		UK	
Avidity	Wt z	BMI z	Wt z	BMI z	Wt z	BMI z	Wt z	BMI z	Wt z	BMI z
Low	−1.42 (1.7)	−0.72 (2.1)	−1.87 (1.3)	−1.09 (1.5)	−1.21 (2.0)	−0.50 (2.5)	−1.49 (0.8)	−1.37 (0.7)	0.19 (0.8)	0.67 (0.5)
Medium	−1.10 (1.5)	−0.49 (1.6)	−1.03 (1.2)	−0.44 (1.4)	−2.06 (1.6)	−1.62 (2.6)	−1.89 (1.1)	−0.92 (1.0)	0.16 (1.2)	0.52 (1.2)
High	−0.82 (1.4)	−0.19 (1.4)	−0.80 (1.3)	−0.33 (1.3)	−1.51 (1.4)	−1.06 (1.9)	−1.44 (1.2)	−0.63 (1.1)	0.40 (1.1)	0.84 (1.2)
*p* nonlinear	0.003	0.03	0.001	0.01	0.20	0.19	0.12	0.10	0.64	0.48
*p* linear	0.001	0.01	0.001	0.01	0.16	0.14	0.17	0.03	0.39	0.34
**Avoidance**										
Low	−1.26 (1.7)	−0.44 (2.0)	−0.94 (1.5)	−0.46 (1.5)	−1.58 (2.1)	−0.74 (2.9)	−1.52 (1.1)	−0.55 (1.1)	0.38 (1.1)	1.00 (1.1)
Medium	−0.68 (1.4)	−0.23 (1.5)	−1.08 (1.2)	−0.54 (1.5)	−0.62 (1.6)	0.04 (2.1)	−1.52 (1.1)	−1.05 (0.9)	0.33 (1.1)	0.59 (1.2)
High	−1.52 (1.5)	−0.81 (1.5)	−1.58 (1.4)	−0.81 (1.4)	−1.95 (1.6)	−1.67 (1.7)	−1.90 (1.3)	−0.90 (1.0)	0.03 (1.2)	0.66 (1.2)
*p* nonlinear	0.001	0.01	0.04	0.45	0.02	0.03	0.31	0.06	0.59	0.48
*p* linear	0.15	0.07	0.02	0.23	0.75	0.25	0.21	0.07	0.37	0.44

## Data Availability

Analyses of the data are still underway, although they can potentially be accessed by enquiry to the first author.

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
