# Peer review of "Development of a Scale to Measure Infant Eating Behaviour Worldwide"

_nutrients, 2021, doi:10.3390/nu13082495_

Round 1
Reviewer 1 Report
The authors write in the first part of the article about the purpose of the program, and what was the purpose of their work in this article? What the authors posted under the "our objectives" section - should be transferred to the Literature Review section summary.
In the research group selection section, complete the information on how many participants took part in the study in each place.
The Discussion and Conclusion sections should be completely redrafted.
The discussion is a comparison of your research results with the results of other researchers - which is difficult to observe in the discussion presented by the authors.
Conclusion: in paragraphs the most important summary elements
The limitations section is also missing.
The presented literature is weak, the most important objection is that there is little of it.
Child malnutrition is a very important issue on which many scientific publications have been written, which the authors could refer to.
Author Response
The authors write in the first part of the article about the purpose of the program, and what was the purpose of their work in this article? What the authors posted under the "our objectives" section - should be transferred to the Literature Review section summary.
DONE
In the research group selection section, complete the information on how many participants took part in the study in each place.
I AM NOT SURE WHAT THE REVIEWER MEANS BY “RESEARCH GROUP SELECTION SECTION”, BUT WE HAVE ADDED THE SUBGROUP NUMBERS TO THE RESULTS SECTION FOR STUDY 2 (THEY WERE ALREADY LISTED FOR STUDY 1) AND TO SUPP TABLE 3
The Discussion and Conclusion sections should be completely redrafted. The discussion is a comparison of your research results with the results of other researchers - which is difficult to observe in the discussion presented by the authors..
The presented literature is weak, the most important objection is that there is little of it.
Child malnutrition is a very important issue on which many scientific publications have been written, which the authors could refer to.
WE HAVE NOT COMPLETELY REWRITTEN THIS, AS THE OTHER REVIEWER FELT IT WAS GOOD, BUT WE HAVE MADE EXTENSIVE ADDITIONS. THERE ARE VERY FEW STUDIES THAT ADDRESS THE SPECIFIC TOPIC OF MEASURING EATING BEHAVIOUR USING A PARENTALLY COMPLETED SCALE IN AN LMIC CONTEXT. HOWEVER, WE HAVE NOW ADDED A WIDER RANGE OF CITATIONS REGARDING EATING BEHAVIOUR IN GENERAL IN THE CONTEXT OF MALNUTRITION
The limitations section is also missing
WE HAD ALREADY MENTIONED A NUMBER OF LIMITATIONS IN LINES 284-297 BUT WE HAVE NOW EDITED IT TO MAKE IT CLEARER THAT THIS A LIMITATIONS SECTION
Conclusion: in paragraphs the most important summary elements
WE HAVE NOW MOVED SOME MATERIAL OUT INTO THE DISCUSSION
Reviewer 2 Report
This well written paper describes the development and validation in different languages of a short questionnaire on eating behaviour rating both avidity and avoidance. Introduction, methods and results are clearly presented. My only remark concerns the possible use of the questionnaire, I would propose that the authors indicate how they envision future use in research? in clinic? Differenciating "normal" from "pathological" eating behaviour?
Author Response
This well written paper describes the development and validation in different languages of a short questionnaire on eating behaviour rating both avidity and avoidance. Introduction, methods and results are clearly presented.
THANK YOU
My only remark concerns the possible use of the questionnaire, I would propose that the authors indicate how they envision future use in research? in clinic? Differenciating "normal" from "pathological" eating behaviour
WE HAVE NOW ADDED A SECTION ON THIS IN THE DISCUSSION
Round 2
Reviewer 1 Report
Dear authors, I am very glad that you took these suggestions into account. With the "limitations" section, I meant to separate it from the whole discussion into a separate item. Same thing for me for strengths. You should do it like this:
4. Discussion
4.1 Strengths
4.2 Limitations
Please mark what new items are added as citations you mentioned - they are not marked in the reference as changes made.
Author Response
We have now added the sections as requested, with subheadings
Most of the new citations were to papers already mentioned in the introduction, but the three entirely new citations were refs 24, 28 and 29:
24: Black, R.E.; Victora, C.G.; Walker, S.P.; Bhutta, Z.A.; Christian, P.; de Onis, M.; Ezzati, M.; Grantham-McGregor, S.; Katz, J.; Martorell, R., et al. Maternal and child undernutrition and overweight in low-income and middle-income countries. The Lancet 2013, 382, 427-451.
28: Moore, A.C.; Akhter, S.; Aboud, F.E. Responsive complementary feeding in rural Bangladesh. Soc.Sci.Med. 2006, 62, 1917-1930.
29: Wright, C.; Garcia, A.; Mutoro, A.; Khan, A.; Milligan, B.; Traynor, O.; Bryant-Waugh, R.; Kimani-Murage, E.; Mayén, V.A. Are Malnourished Children Hungry? Use of the International Complementary Feeding Assessment Tool (ICFET) to Describe Diet and Eating Behavior. Current Developments in Nutrition 2020, 4, 924-924, doi:10.1093/cdn/nzaa053_129.